# Counteracting Roles of Lipidic Aldehydes and Phenolic Antioxidants on Soy Protein Oxidation Defined by a Chemometric Survey of Solvent and Mechanically Extracted Soybean Meals

**DOI:** 10.3390/antiox12071419

**Published:** 2023-07-13

**Authors:** Junwei Zhang, Pedro E. Urriola, Seth L. Naeve, Gerald C. Shurson, Chi Chen

**Affiliations:** 1Department of Food Science and Nutrition, University of Minnesota, St. Paul, MN 55108, USA; zhan8011@umn.edu; 2Department of Animal Science, University of Minnesota, St. Paul, MN 55108, USA; urrio001@umn.edu; 3Department of Agronomy and Plant Genetics, University of Minnesota, St. Paul, MN 55108, USA; naeve002@umn.edu

**Keywords:** aldehydes, antioxidant, isoflavones, lipid oxidation, protein oxidation, soybean meal

## Abstract

Soybean meal (SBM) is a premier source of protein for feeding food-producing animals. However, its nutritional value can be compromised by protein oxidation. In this study, a total of 54 sources of solvent extracted SBM (SSBM) and eight sources of mechanically extracted SBM (MSBM), collected from different commercial producers and geographic locations in the United States during the years 2020 and 2021, were examined by chemometric analysis to determine the extent of protein oxidation and its correlation with soybean oil extraction methods and non-protein components. The results showed substantial differences between SSBM and MSBM in the proximate analysis composition, protein carbonyl content, lipidic aldehydes, and antioxidants, as well as subtle differences between 2020 SSBM and 2021 SSBM samples in protein oxidation and moisture content. Correlation analysis further showed positive correlations between protein carbonyl content and multiple lipid parameters, including the ether extract, *p*-anisidine value, individual aldehydes, and total aldehydes. Among the antioxidants in SBM, negative correlations with protein carbonyl content were observed for total phenolic content and isoflavone glycoside concentrations, but not for Trolox equivalent antioxidant capacity (TEAC), α-tocopherol, and γ-tocopherol. Overall, soybean oil extraction methods, together with other factors such as enzyme treatment and environmental conditions, can significantly affect the proximate analysis composition, the protein and lipid oxidation status, and the antioxidant profile of SBM. Lipidic aldehydes and phenolic antioxidants play counteracting roles in the oxidation of soy protein. The range of protein carbonyl content measured in this study could serve as a reference to evaluate the protein quality of SBM from various sources used in animal feed.

## 1. Introduction

Soybean meal (SBM) is the by-product of soybean oil extraction, with annual production of 257 million metric tons globally in 2022 [1]. SBM is the primary source of protein in the diets of many food-producing animals because of its favorable attributes, including high crude protein content, a balanced amino acid profile, high amino acid digestibility, palatability, and wide availability [2]. Hexane-based processing is the most widely used method to extract oil from soybeans, yielding solvent-extracted SBM (SSBM). Alternatively, mechanically extracted SBM (MSBM) can be produced by applying continuous mechanical screw processes or expellers to extract soybean oil. In general, solvent extraction is more efficient than mechanical extraction in recovering oil, resulting in as little as 0.5% residual oil in SSBM but approximately 5% to 8% residual oil in MSBM [3,4]. Despite this difference, modest amounts of soybean are still processed by mechanical extraction for reasons including lower initial capital costs and the avoidance of solvent usage [3].

In recent years, the use of enzyme treatment to enhance the feeding value of SBM has also attracted attention, since enzyme-treated SBM contains greater crude protein, amino acid, and small peptide content and lower concentrations of anti-nutritional factors than conventional SSBM [5]. Studies have shown that feeding diets containing enzyme-treated SBM to weaned pigs improves the dietary protein digestibility, growth performance, antioxidant index, immune function, and gut health [5,6,7]. Currently, there is a very limited supply available of enzyme-treated SBM produced in the United States for more comprehensive comparisons.

Inappropriate processing and storage can cause the oxidation of SSBM and MSBM, compromising their nutritional value. Feeding oxidized SBM to livestock and poultry has been shown to reduce the digestibility of amino acids (e.g., lysine), disrupt the antioxidant status, and impair growth performance [8,9,10]. Unfavorable physical conditions, including thermal exposure (e.g., toasting) and pressure (extrusion), along with pro-oxidant chemical factors including moisture, water activity, and oil content, can cause significant protein and lipid oxidation, while natural soy phytochemicals such as isoflavones, phenolic acids, and tocopherols can function as antioxidants [11,12]. Normally, controlled heating is purposely used in SBM processing to denature antinutritional native soy proteins, especially trypsin inhibitors, in raw soybeans to improve their nutritional value [13,14]. However, excessive heating oxidizes the proteins in SBM and leads to the peroxidation of soybean oil [9,10,15,16]. In addition, moisture and water activity are commonly used as indicators to predict the susceptibility and resistance of feed ingredients to spoilage during storage, because excessive free water promotes microbial growth, nonenzymatic spontaneous reactions, and enzyme activity [17,18,19]. The consequences of these pro-oxidation events are the peroxidation of unsaturated fatty acids and the formation of lipid oxidation products (LOPs) [15,16]. Among the diverse LOPs, reactive aldehydes can covalently bond with amino and sulfhydryl groups of amino acids, forming carbonylated soy proteins with decreased solubility and digestibility [20,21].

Comparative surveys on the nutrient and chemical compositions of sources of SBM are routinely conducted to determine their quality and to examine the influences of climate, management practices, and processing conditions [22,23]. However, limited data are available on the extent of protein and lipid oxidation in commercial SBM sources, types of SBM, and associated contributing factors. Therefore, the objectives of this study were to measure the concentrations of protein carbonyl, lipidic aldehydes, and antioxidants in a total of 62 MSBM and SSBM samples produced in years 2020 and 2021 from various geographical locations in the United States. The effects of different processing methods on soy protein oxidation were further evaluated by determining correlations between oxidation measures and pro- and anti-oxidation factors.

## 2. Materials and Methods

### 2.1. Soybean Meal Samples

A total of 62 SBM samples were generously provided by commercial soybean processors in the United States, including 38 SSBM from the year 2020 (2020 SSBM), 16 SSBM from the year 2021 (2021 SSBM), and 8 MSBM from the year 2021 (2021 MSBM). The 2021 SSBM contained one HP300 sample (Hamlet Protein, Findlay, OH, USA), which was produced by a proprietary enzyme treatment of conventional SSBM. All SBM samples were ground and sieved through a #18 mesh, and then sealed in polypropylene bags and stored at 4 °C before analysis.

### 2.2. Chemicals and Reagents

The chemicals and reagents used in sample preparation, chemical analysis, LC-MS analysis, structural confirmation, and quantification are listed in Appendix A.

### 2.3. Proximate Analysis

Samples of SBM were analyzed for moisture, crude protein, ether extract, crude fiber, and ash using the Association of Official Agricultural Chemists (AOAC) procedures (Appendix A) by the University of Missouri Agricultural Experiment Station Chemical Laboratories.

### 2.4. Water Activity

The water activity of SBM samples was measured in duplicate using an AQUA LAB CX-2 water activity meter (Decagon Devices, Pullman, WA, USA) [24].

### 2.5. Protein Carbonyl Content

The protein carbonyl content of SBM was analyzed using a modified 2,4-dinitrophenylhydrazone (DNPH)-based method [25]. Soy protein was extracted by mixing 0.3 g of SBM with 5 mL of water, followed by 30-min sonication, 10-min vortexing, and 10-min centrifugation at 18,000× *g.* After passing the supernatant through a 0.22 µm syringe filter, 0.5 mL of the filtrate was mixed with 0.1 mL of 10 mM DNPH in 2 M HCl in a 1.5 mL Eppendorf tube by vortexing for 60 min, followed by the addition of 0.6 mL of 20% trichloroacetic acid (TCA) solution, chilling on ice for 20 min, and 10-min centrifugation at 18,000× *g*. The precipitated protein pellet was washed three times by 5-min vortexing with 1 mL of ethanol-ethyl acetate (1:1), followed by 10-min centrifugation at 18,000× *g* and decanting the supernatant after each wash. The protein pellet was dried with nitrogen and redissolved with 0.5 mL of 6 M guanidine hydrochloride by vortexing for 20 min. The absorbance of the protein solution (As) was measured at 370 nm using a spectrophotometer. The absorbance of the blank solution of 0.1 mL of 2 M HCl without DNPH (Ab) was also measured. The soluble protein concentration was determined using a BCA Protein Assay Kit (Thermo Fisher Scientific, Waltham, MA, USA), and the protein carbonyl content was calculated with the following equation:$$Carbonyl\;content\left(µmol/gprotein\right)=\frac{(As-Ab)/\varepsilon }{Soluble\;protein\;concentration}$$
in which ε is the molar absorptivity coefficient of 22,000/M/cm.

### 2.6. p-Anisidine Value (p-AV)

The *p*-AV was measured according to the Official Method Cd 18–90 from the American Oil Chemists Society (AOCS). Briefly, 2 g of SBM was mixed with 5 mL of isooctane, followed by 30-min vortexing and 10-min centrifugation at 18,000× *g*, and the supernatant was collected for the *p*-AV analysis according to the official method.

### 2.7. Aldehydes

Aldehydes in the SBM samples were extracted with isooctane and then analyzed based on a previously established method [16]. First, 2 g of SBM was mixed with 5 mL of isooctane, followed by 30-min vortexing and 2-min centrifugation at 1500× *g*. This extraction was conducted three times, and all the supernatants were collected and dried under a stream of nitrogen, obtaining residual oil containing aldehydes. Then, 2 µL of the oil was used for 2-hydrazinoquinoline (HQ)-based derivatization [16], and a 5 µL aliquot was injected into an ultraperformance liquid chromatography (UPLC) system equipped with a BEH C18 column (Waters, Milford, MA, USA) using a mobile phase gradient for separation (Appendix A). The liquid chromatography (LC) eluant was introduced into a Xevo-G2-S quadrupole time of flight mass spectrometer (QTOFMS) system (Waters) for accurate mass measurement and ion counting. For accurate mass measurement, the mass spectrometer was calibrated with sodium formate solution with a mass-to-charge ratio (*m*/*z*) of 50–1200 and monitored by the intermittent injection of lock mass leucine enkephalin ([M + H]^+^ = *m*/*z* 556.2771) in real time. Mass chromatograms and mass spectral data were acquired and processed using the MassLynx^TM^ software V4.2 (Waters) in centroided format. The chemical identities of compounds of interest were identified by accurate mass-based elemental composition analysis, MSMS fragmentation, and comparisons with authentic standards. Individual compound concentrations were determined by calculating the ratio between the peak area of compound and the peak area of the internal standard and fitting with a standard curve using the Quanlynx^TM^ software V4.2 (Waters).

### 2.8. Trolox Equivalent Antioxidant Capacity (TEAC)

The antioxidant compounds in SBM samples were extracted and subsequently used for the TEAC analysis [26]. Briefly, 50 mg of SBM was mixed with 500 µL of methanol, followed by 15-min vortexing, 15-min sonication, and 10-min centrifugation at 18,000× *g*. The extraction was conducted three times, and all the supernatants were dried with nitrogen. The extract was redissolved with 200 µL of 60% aqueous methanol. Afterwards, (2,2’-azino-bis (3-ethylbenzothiazoline-6-sulphonic acid)) diammonium salt (ABTS) stock solution (7 mM in water) was reacted with 2.45 mM potassium persulfate, and then kept in the dark for 12–16 h to generate an ABTS^•+^ solution. The ABTS^•+^ solution was diluted with ethanol to reach absorbance of 0.70 ± 0.02 at 734 nm before use. Next, 500 μL of diluted ABTS^•+^ solution was mixed with 20 μL of the extract solution. After 6-min incubation at room temperature, the absorbance was measured at 734 nm. Finally, 6-hydroxy-2,5,7,8-tetramethylchroman-2-carboxylic acid (Trolox) was used for the standard curve.

### 2.9. Tocopherols

The α- and γ-tocopherol concentrations in SBM samples were analyzed based on a previously established method [27]. Briefly, 100 mg of SBM was mixed with 1 mL of methanol, followed by 15-min vortexing, 15-min sonication, and 10-min centrifugation at 18,000× g. The extraction was conducted three times, and all the supernatants were dried under a stream of nitrogen. Then, the extract was reconstituted with 200 µL of methanol containing 5 µg/mL tripentadecanoin (internal standard), and a 5 μL aliquot of the extract solution was injected into the same UPLC-MS system as described for the aldehyde analysis but using a different mobile phase gradient for separation (Appendix A).

### 2.10. Total Phenolic Content

The total phenolic content of SBM samples was analyzed using the Folin–Ciocalteu method [26]. A 40 µL aliquot of the extract solution from the TEAC assay was mixed with 460 µL of water and 50 µL of Folin–Ciocalteu reagent. After 3 min, 100 µL of 35% sodium carbonate solution and 350 µL of water were added into the mixture, followed by 60-min incubation at ambient temperature. The absorbance was measured at 725 nm using a spectrometer. Caffeic acid was dissolved in 60% methanol and used for the standard curve.

### 2.11. Isoflavones

The concentrations of isoflavones (genistin, genistein, daidzin, and daidzein) were analyzed based on a previously established method [28]. Briefly, 60 mg of SBM was mixed with 600 µL of 70% methanol containing 1 µM sulfadimethoxine (internal standard), followed by two rounds of 15-min vortexing and 15-min sonication. After 15-min centrifugation at 18,000× *g*, the supernatant was collected for isoflavone quantification using the same UPLC-MS system as described for the aldehyde analysis but using a different mobile phase gradient for separation (Appendix A).

### 2.12. Statistical Analysis

Data were analyzed using GraphPad Prism 9.30 (GraphPad Software, San Diego, CA, USA), and statistical analysis was conducted with standard one-way ANOVA and Tukey’s multiple comparisons tests. Correlations were analyzed by two-tailed Pearson correlation analysis. Statistically significant differences were noted when *p* < 0.05, and 0.05 ≤ *p* ≤ 0.10 was considered a statistical trend.

## 3. Results

### 3.1. Proximate Analysis and Water Activity of SSBM and MSBM

The proximate analysis showed that SSBM samples produced during 2020 and 2021 had greater crude protein and ash content (Figure 1A,B) but less ether extract and crude fiber (Figure 1C,D) compared with 2021 MSBM samples. Both 2020 and 2021 SSBM samples had greater moisture content than 2021 MSBM samples (Figure 1E), which was consistent with the greater water activity in SSBM samples (Figure 1F) compared with MSBM samples. The greater moisture content in the 2020 SSBM samples compared with 2021 SSBM samples was unexpected (Figure 1E). In addition, the HP300 sample deviated from the other conventional 2021 SSBM samples in its greater crude protein (Figure 1A), reduced moisture content (Figure 1E), and lower water activity (Figure 1F).

### 3.2. Protein Oxidation of SSBM and MSBM

The protein oxidation of SBM samples was evaluated by determining their protein carbonyl content. The results showed that both 2020 and 2021 SSBM samples had lower carbonyl concentrations than the 2021 MSBM samples (Figure 2). Moreover, the 2021 SSBM samples had greater protein carbonyl content than the 2020 SSBM samples (Figure 2).

### 3.3. Lipid Oxidation of SSBM and MSBM

The lipid oxidation of SBM samples was first evaluated by determining their *p*-AVs. The results showed that the *p*-AVs of 2020 and 2021 SSBM samples were comparable, but both had smaller *p*-AVs than observed in the 2021 MSBM samples (Figure 3A). Within the 2021 SSBM samples, the HP300 sample had a greater *p*-AV than the other conventional SSBM samples (Figure 3A). Subsequent analysis of individual aldehydes (from C6–C10) showed variation in their distribution profiles across different sample groups and within the same sample group. Among these aldehydes, 2-heptenal was the most abundant aldehyde in all SBM samples (Figure 3D). The 2021 MSBM samples contained higher concentrations of 2-hexenal, 2-heptenal, octanal, 2-octenal, nonanal, 2,4-nonadienal, and 2-decenal than the 2020 SSBM samples (Figure 3B,D–G,I,J) and greater concentrations of octanal, nonanal, and 2,4-nonadienal than the 2021 SSBM samples (Figure 3E,G,I). Between the two groups of SSBM samples, 2021 SSBM samples had more 2,4-heptadienal (Figure 3C) but less nonanal (Figure 3G) than the 2020 SSBM samples. No differences in 2-nonenal concentrations were observed among the three SBM groups evaluated (Figure 3H). Total aldehydes were greater in the 2021 MSBM samples than in the 2020 SSBM samples but not for the 2021 SSBM samples (Figure 3K).

### 3.4. Total Antioxidant Capacity and Tocopherol Concentrations of SSBM and MSBM

The antioxidant capacity of SSBM and MSBM samples was evaluated using the TEAC assay to calculate a Trolox-equivalent value. The results showed that the 2020 and 2021 SSBM samples had comparable TEAC values but greater antioxidant capacity than the 2021 MSBM samples (Figure 4A). However, the subsequent analysis of tocopherols showed that both 2020 and 2021 SSBM samples had less α-tocopherol and γ-tocopherol than the 2021 MSBM samples (Figure 4B,C), indicating that other antioxidants besides tocopherols contributed to the increased antioxidant capacity of SSBM.

### 3.5. Total Phenolic and Isoflavone Concentrations of SSBM and MSBM

The total phenolic content of SBM samples was evaluated by the Folin–Ciocalteu assay. The results showed that the 2020 and 2021 SSBM samples had comparable total phenolic content, but they both contained more phenolics than the 2021 MSBM samples (Figure 5A). Isoflavones are the predominant and unique phenolic compounds in SBM, and the following were characterized by LC-MS analysis in both SSBM and MSBM samples: daidzin (I), glycitin (II), genistin (III), 6″-*O*-malonyldaidzin (IV), 6″-*O*-acetyldaidzin (V), daidzein (VI), 6″-*O*-acetylgenistin (VII), and genistein (VIII) (Figure 5B, Table 1). The concentrations of daidzin, genistin, daidzein, and genistein in SBM were quantified using authentic standards, and results indicated that daidzin and genistin, two glycosides, were far more abundant than daidzein and genistein, their respective aglycones (Figure 5C–F). The concentrations of daidzin and genistin in both 2020 and 2021 SSBM samples were greater than those of the 2021 MSBM (Figure 5C,D). In contrast, 2020 SSBM samples had less genistein than 2021 MSBM samples (Figure 5F), while the concentrations of daidzein were comparable across the three sample groups (Figure 5E). In addition, the HP300 sample contained greater daidzein and genistein concentrations than conventional 2021 SSBM samples (Figure 5E,F).

### 3.6. Correlations of Proximate Analysis Composition, Lipid Oxidation, and Antioxidants with Protein Oxidation in SBM

Individual parameters of the proximate analysis, lipid oxidation, and antioxidants were correlated with the protein carbonyl content for their potential associations and contributions to protein oxidation in SSBM and MSBM samples. Among the proximate analysis parameters, crude fiber and ether extract were positively correlated with the protein carbonyl content of SBM, while moisture and water activity were negatively correlated (Figure 6). The parameters of lipid oxidation, including *p*-AV, total aldehydes, and individual aldehydes, were all positively correlated with protein carbonyl content (Figure 6). Among the measured antioxidants, negative correlations with protein carbonyl content were observed for total phenolics and daidzin, but not for daidzein, TEAC, α-tocopherol, and γ-tocopherol (Figure 6). In addition, genistin tended to be negatively correlated with protein carbonyl content (*p* = 0.063), while genistein, which is a corresponding isoflavone aglycone, had a positive correlation with protein oxidation (Figure 6).

## 4. Discussion

Protein and lipid oxidation are detrimental characteristics that reduce the nutri-physiological value of feed ingredients because oxidized proteins and lipids, such as carbonylated protein, crosslinked peptides, and lipidic aldehydes, in feed ingredients can lead to oxidative stress, impaired protein and lipid digestibility, and compromised intestinal function in food-producing animals [9,10,29]. Therefore, measuring the oxidation levels of feed ingredients, together with conventional nutrient composition analysis, can provide a more complete evaluation of their nutri-physiological value, facilitating the practice of precision animal nutrition [30]. Results from our chemometric survey of multiple and diverse sources of commercial SSBM and MSBM samples provide a comprehensive overview of the oxidation status of SBM from industrial production in the United States. More importantly, these data demonstrate the potential associations and contributions of the processing methods, and the presence and concentrations of prooxidants, antioxidants, and other physicochemical factors associated with the protein oxidation of SBM, through correlation analyses (Figure 7).

### 4.1. Protein Oxidation Status in SSBM and MSBM

Protein oxidation encompasses diverse chemical modifications on amino acid side chains and the peptide backbone [31]. The formation of protein carbonyls is a common type of protein oxidation, derived from the direct oxidation of amino acid side chains by reactive radicals or the binding of amino acid side chains with reactive carbonyls, such as the aldehydes from lipid peroxidation, through the Michael addition reaction or the formation of Schiff bases [32]. Compared with other protein-related quality indicators of SBM, such as trypsin inhibitor activity, urease activity, KOH solubility, the protein dispersibility index, and reactive lysine, published data on the protein carbonyl content in SBM have been limited to only a few reports with small sample sizes [9,10,33]. Based on our literature search, the current study is the first comprehensive survey of the protein carbonyl content in commercial SSBM and MSBM sources and it showed that the majority of the 62 tested SBM samples had protein carbonyl content within the range of 3–12 µmol/g protein, with averages of 5.6 µmol/g protein for SSBM and 9.2 µmol/g protein for MSBM. Considering that the protein carbonyl content has been used in monitoring the protein quality of other foods and feed ingredients [34], our results could serve as a reference in practice in evaluating and comparing the protein quality of commercial SBM sources. This recommendation is further supported by the observations that feeding heated SBM with protein carbonyl content within the range of 8.5–12 µmol/g protein has been shown to impair the growth performance, digestive function, and antioxidant status of young broilers and laying hens [9,10,33]. Therefore, more animal-feeding-based nutritional studies are needed to define the associations between protein carbonyl content and production performance in different animal species and to determine the threshold of toxicological concern for protein carbonyls in the dietary exposure of oxidized SBM.

Processing and storage are two external factors that may affect the protein oxidation status of SBM. Results from a previous study showed that the protein in defatted soy flour was chemically stable after 250 days in storage at 25, 4, and −20 °C [35]. Low moisture and water activity in SBM, as shown in the current study, also favor stability in long-term storage. Therefore, storage under ambient conditions is not likely to be a significant contributor to protein carbonylation in SBM. In contrast, the influences of the processing method on protein carbonyl production in SBM are evident in the current study—the carbonyl concentrations in MSBM were greater than those in conventional SSBM. This phenomenon is related to exposure to different thermal treatments in the process used to extract soybean oil and subsequently produce SSBM and MSBM. Previous investigations have shown that thermal treatment causes significant increases in the protein carbonyl content of SBM [9,10]. The process used to produce SSBM involves mild heat exposure during solvent extraction (e.g., toasting), but high pressure and heat occur with continuous screw pressing during the processing of MSBM [4]. Therefore, the significant difference in protein carbonyl content between SSBM and MSBM can likely be attributed to processing conditions rather than storage. In addition, mechanical extraction is commonly used for the small-volume production of soybean oil and involves more variation in the types of extraction equipment (e.g., extruder or expeller) and processing parameters (e.g., extrusion temperature) than solvent extraction [3,36]. These factors may account for the greater variation in the protein carbonyl concentrations of MSBM (Figure 2), which may result in variable animal growth performance and amino acid digestibility responses when animals are fed MSBM [36,37]. Moreover, the observation of rather high protein carbonyl content (9.3 µmol/g protein) in the HP300 sample, which was produced by the proprietary enzyme treatment of conventional SSBM and post-treatment heating [38,39], also highlights the impact of processing.

Another interesting observation is that the 2021 SSBM samples had greater protein carbonyl content than the 2020 SSBM samples, even though they were both produced by solvent extraction and had comparable proximate analyses. According to the National Oceanic and Atmospheric Administration (NOAA), 2021 was the fourth-warmest year in the 127-year period of NOAA’s record (https://www.ncei.noaa.gov/news/national-climate-202112, accessed on 17 May 2023) and had above-average temperatures in the soybean production areas of the United States (Appendix A). Together with the lower moisture content in 2021 SSBM samples, these observations may imply an effect of climate during soybean production on the protein oxidation of SBM.

### 4.2. Correlations of Lipid Oxidation with Protein Oxidation in SSBM and MSBM

Lipids, mainly unsaturated fatty acids, are the sources of lipidic aldehydes for protein carbonylation reactions. Mechanical extraction is not as efficient as solvent extraction in the recovery of soybean oil, leaving more residual oil in MSBM. The residual oil is subject to oxidation under the same conditions that promote the oxidation of soy protein, including thermal treatment and oxygen, resulting in elevated *p*-AV, peroxide values, and unsaturated aldehydes [15,16]. Many reactive aldehydes, including 2,4-decadienal, malondialdehyde, and 4-hydroxy-2-nonenal, can react with the side chains of amino acids (e.g., lysine) to form protein carbonyls, compromising the physicochemical properties (e.g., solubility) and nutritional value (e.g., digestibility) of the affected proteins [20,40]. Extensive evidence supports the roles of lipid oxidation in the oxidation of soy protein. For example, full-fat soy flour was more prone to protein carbonylation under the same storage conditions as defatted soy flour [35]. In addition, a soy protein isolate incubated with highly oxidized soybean oil and fish oil with greater *p*-AV produced more protein carbonyls than fresh oils with lower oxidation levels [21]. Therefore, it is not surprising that we observed strong positive correlations in this study between protein carbonyls and multiple lipid parameters, including the ether extract, *p*-AV, individual aldehydes, and total aldehydes (Figure 7). The management of the residual oil content and its oxidation could play a key role in the processing and storage practices to control protein oxidation in SBM.

### 4.3. Correlations of Antioxidants with Protein Oxidation in SSBM and MSBM

Soybean meal contains several bioactive phytochemicals with antioxidant properties [12,41], including diverse phenolic compounds and tocopherols, but the correlations between these antioxidants and protein oxidation were not consistent in this study (Figure 7). First, TEAC, α-tocopherol, and γ-tocopherol were not significantly correlated with protein carbonyl content (Figure 6). In fact, SSBM samples had less α- and γ-tocopherol than MSBM samples, which was potentially due to the lower residual oil content in SSBM, even though they had higher TEAC values (Figure 4). These observations indicate that tocopherols are not the major contributors to the total antioxidant activity of SBM. Secondly, a negative correlation between total phenolic content and protein carbonyl content was observed. Phenolic acids, flavonoids, anthocyanin, and tannic acids are the major phenolic components in soybeans and SBM [12,42]. Among the examined isoflavones, daidzin and genistin, which were the two dominant isoflavone glycosides in SBM samples, had negative correlations with protein oxidation. In contrast, genistein, which is a corresponding isoflavone aglycone, had a positive correlation with protein carbonyl content (Figure 6). These differences in correlation responses may be attributed to the different oil extraction methods used to produce these types of SBM. The extrusion process in mechanical extraction, which increases both lipid and protein oxidation, may also promote the de-esterification of isoflavone glycosides to their respective aglycones [43].

### 4.4. Correlations of Proximate Analysis Components with Protein Oxidation in SBM

The observed correlations of protein oxidation with SBM proximate analysis components, including the positive correlations with the ether extract and crude fiber and the negative correlations with moisture and water activity (Figure 6), were associated more with SBM processing, indicating that they are not necessarily contributing factors to protein oxidation. Soybean oil extraction methods vary in procedures and extraction efficiency, leading to marked differences in the proximate analysis components of SBM (Figure 1). The observed lower crude protein of MSBM samples compared with that of SSBM samples, together with their greater crude fiber, was caused by the absence of a dehulling process in mechanical extraction [4]. The ether extract levels of the MSBM and SSBM samples evaluated in this study were comparable with the reported values (approximately 1% residual oil for SSBM and 5–8% for MSBM) in the database of the National Animal Nutrition Program (https://animalnutrition.org/, accessed on 27 March 2023), confirming that soybean oil is more completely recovered by hexane extraction than mechanical extraction [4,44]. The differences in soybean oil extraction processes also lead to greater moisture content and water activity in SSBM compared with MSBM. Soybeans typically have moisture content of 13% when harvested. During the solvent extraction process used to produce SSBM, the desolventizing–toasting process removes hexane, with limited loss of water. In contrast, the pressure and heat from continuous hard screw pressing in mechanical extraction evaporate water, resulting in less moisture remaining in MSBM [3].

## 5. Conclusions

This comprehensive chemometric survey of 62 commercial SSBM and MSBM samples in the United States showed substantial differences between SSBM and MSBM in the proximate analysis composition, protein carbonyl content, lipidic aldehydes, and antioxidants, as well as subtle differences between 2020 SSBM and 2021 SSBM samples in protein oxidation and moisture content. The correlation analysis revealed that protein oxidation was likely caused by lipidic aldehydes, while discriminately counteracted by soy antioxidants. The observed range of protein carbonyl content in this study may serve as a useful reference value in evaluating the protein quality of SBM used in animal feed. More animal-feeding-based nutritional studies are needed to examine the influences of protein oxidation on SBM-fed food-producing animals.

## Figures and Tables

**Figure 1 antioxidants-12-01419-f001:**
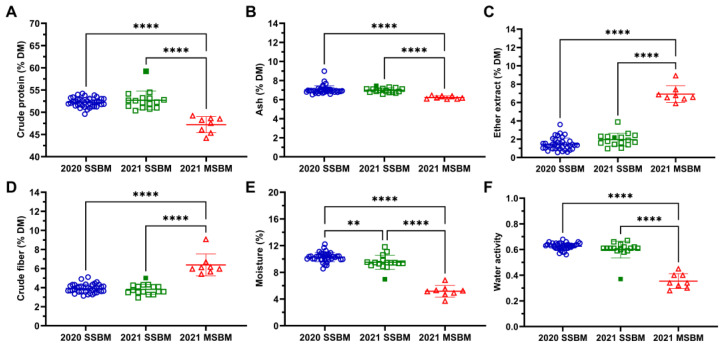
Proximate analysis and water activity of SSBM and MSBM samples. (**A**) crude protein, (**B**) ash, (**C**) ether extract, (**D**) crude fiber, (**E**) moisture, and (**F**) water activity. The HP300 sample in the 2021 SSBM is shown as a solid green square. Numerical values of proximate analysis and water activity are listed in Appendix A. Data are presented as mean ± SD. **, *p* < 0.01; ****, *p* < 0.0001 from one-way ANOVA and Tukey’s multiple comparison.

**Figure 2 antioxidants-12-01419-f002:**
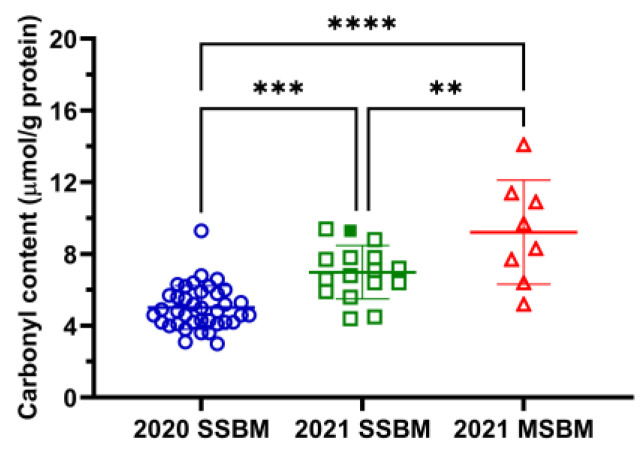
Protein carbonyl content of SSBM and MSBM samples. The HP300 sample in the 2021 SSBM is shown as a solid green square. Numerical values of protein carbonyl content are listed in Appendix A. Data are presented as mean ± SD. **, *p* < 0.01; ***, *p* < 0.001; ****, *p* < 0.0001 from one-way ANOVA and Tukey’s multiple comparison.

**Figure 3 antioxidants-12-01419-f003:**
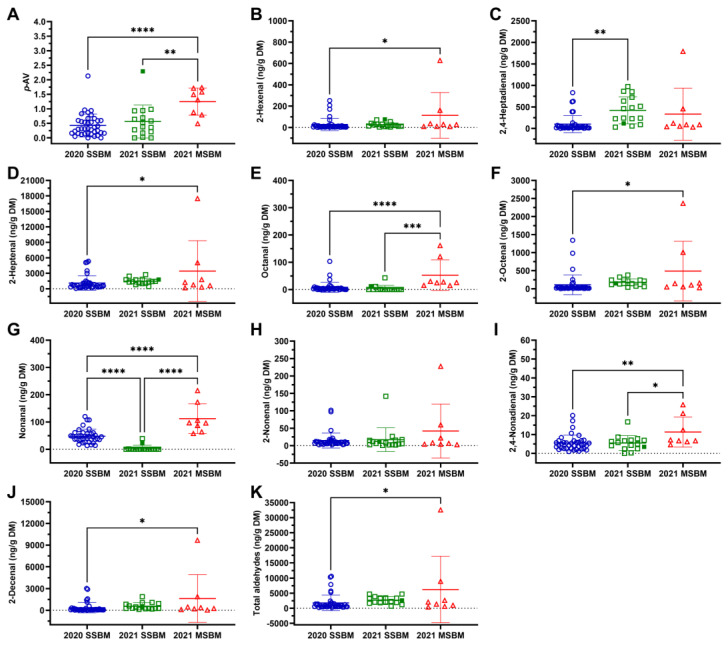
Lipid oxidation of SSBM and MSBM samples. (**A**) *p*-AV, (**B**) 2-hexenal, (**C**) 2,4-heptadienal, (**D**) 2-heptenal, (**E**) octanal, (**F**) 2-octenal, (**G**) nonanal, (**H**) 2-nonenal, (**I**) 2,4-nonadienal, (**J**) 2-decenal, and (**K**) total aldehydes. The HP300 sample in the 2021 SSBM is shown as a solid green square. Numerical values of *p*-AV and aldehydes are listed in Appendix A. Data are presented as mean ± SD. *, *p* < 0.05; **, *p* < 0.01; ***, *p* < 0.001; ****, *p* < 0.0001 from one-way ANOVA and Tukey’s multiple comparison.

**Figure 4 antioxidants-12-01419-f004:**
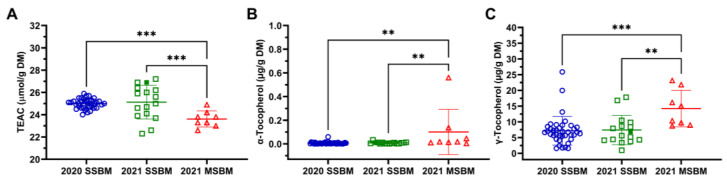
Total antioxidant capacity and tocopherol content of SSBM and MSBM samples. (**A**) TEAC, (**B**) α-tocopherol, and (**C**) γ-tocopherol. The HP300 sample in the 2021 SSBM is shown as a solid green square. Numerical values of TEAC and tocopherols are listed in Appendix A. Data are presented as mean ± SD. **, *p* < 0.01; ***, *p* < 0.001 from one-way ANOVA and Tukey’s multiple comparison.

**Figure 5 antioxidants-12-01419-f005:**
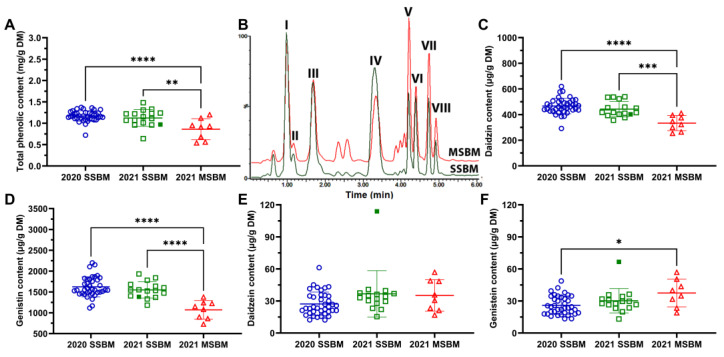
Total phenolic content and isoflavone concentrations of SSBM and MSBM samples. (**A**) Total phenolic content, (**B**) representative LC-MS chromatograms of MSBM and SSBM (peaks of major isoflavones are labeled as I–VIII), (**C**) daidzin (I), (**D**) genistin (III), (**E**) daidzein (VI), and (**F**) genistein (VIII). The HP300 sample in the 2021 SSBM is shown as a solid green square. Numerical values of total phenolic content and isoflavones are listed in Appendix A. Data are presented as mean ± SD. *, *p* < 0.05; **, *p* < 0.01; ***, *p* < 0.001; ****, *p* < 0.0001 from one-way ANOVA and Tukey’s multiple comparison.

**Figure 6 antioxidants-12-01419-f006:**
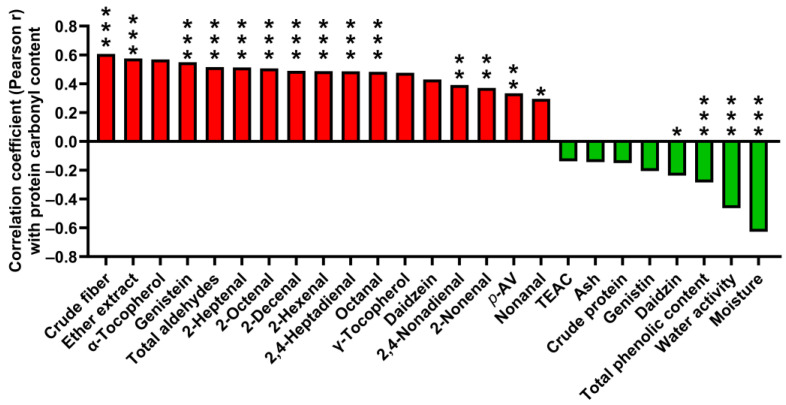
Pearson correlation coefficients of individual parameters of proximate analysis, lipid oxidation, and antioxidants with protein oxidation. *, *p* < 0.05; **, *p* < 0.01; ***, *p* < 0.001.

**Figure 7 antioxidants-12-01419-f007:**
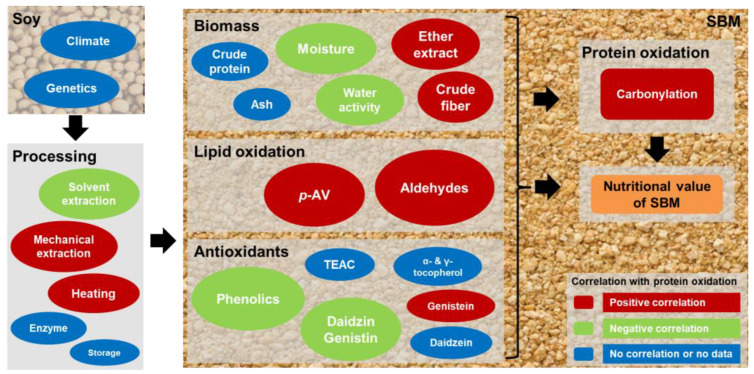
Summary of contributing factors to protein oxidation of SBM and their correlations with protein carbonyls. Red color indicates positive correlation or potential promoting effect on protein oxidation; green color indicates negative correlation or potential preventive effect against protein oxidation; blue color indicates no correlation or a lack of data.

**Table 1 antioxidants-12-01419-t001:** Major isoflavones identified in the LC-MS analysis of SSBM and MSBM samples.

Peak ID	RT (min)	Identity	Formula	Mode of Ion Detection	*m/z* of Detection	Δppm	Database ID
I	0.98	Daidzin *	C_21_H_20_O_9_	M+H	417.1180	0	HMDB0033991
II	1.17	Glycitin #	C_22_H_22_O_10_	M+H	447.1286	0	HMDB0002219
III	1.69	Genistin *	C_21_H_20_O_10_	M+H	433.1129	1	HMDB0033988
IV	3.30	6″-*O*-Malonyldaidzin #	C_24_H_22_O_12_	M+H	503.1184	1	HMDB0041263
V	4.19	6″-*O*-Acetyldaidzin #	C_23_H_22_O_10_	M+H	459.1286	1	HMDB0030689
VI	4.38	Daidzein *	C_15_H_10_O_4_	M+H	255.0652	4	HMDB0003312
VII	4.73	6″-*O*-Acetylgenistin #	C_23_H_22_O_11_	M+H	475.1235	1	HMDB0029528
VIII	4.91	Genistein *	C_15_H_10_O_5_	M+H	271.0601	4	HMDB0003217

* confirmed by authentic standards; # determined by both elemental composition analysis and literature review [28]; Δppm indicates the deviation of the measured mass from the exact mass of the identified compound in parts per million (ppm). The Human Metabolome Database (HMDB, https://hmdb.ca, accessed on 13 May 2023) was used for the database search.

## Data Availability

The chemometric and statistically analyzed data are contained within the article and Appendix A. The raw data of the chromatographic and spectrometric analyses can be requested from the corresponding authors.

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
