# Peer review of "Counteracting Roles of Lipidic Aldehydes and Phenolic Antioxidants on Soy Protein Oxidation Defined by a Chemometric Survey of Solvent and Mechanically Extracted Soybean Meals"

_antioxidants, 2023, doi:10.3390/antiox12071419_

Round 1
Reviewer 1 Report
Dear Authors,
I attach comments in the file.
Yours sincerely,
Reviewer

Author Response
--- Responses to Reviewer 1’s comments
- “Lines 99,100. What was the size of the sample in this test? Was it some method used to measure water activity?”
Reply: The analysis does not require exact sample weight if the moisture released from the sample can reach equilibrium within the sample chamber of the analyzer. In this study, we used 1-2 g of samples for our Aqua Lab CX-2 water activity meter. The method is cited in reference [23] (Line 101), and also described in the manual of Aqua Lab CX-2 water activity meter used in this study (https://usermanual.wiki/Document/AquaLabCX2v3.3724633807).
- “Line 118. There should be a number in parentheses of this relationship and a possible reference to literature, if the authors used it.”
Reply: The method and equation (Line 119) used for the protein carbonyl content are from the reference [25], which has been cited (Line 104).
- “Figs 1 - 6. X-axis description is missing.”
Reply: The X-axis of Figs. 1-6 all contain 3 groups of samples (2020 SSBM, 2021 SSBM, and 2021 MSBM), which are self-explanatory. Therefore, an additional description, such as “Sample groups”, is not be necessary.
- “In the methodological section in section 2.12, the authors mentioned that they performed statistical analysis using standard one-way ANOVA and Tukey's multiple comparison tests and that correlation were analyzed by two-tailed Pearson correlation analysis. However, in the part concerning the research results and in the discussion there are no results and no reference to the results of the ANOVA and Tukey's multiple comparisons tests. There is no information on this subject and no presentation of the results of these tests. This needs supplementing. The Results section presents the Pearson correlation coefficient results.”
Reply: Thanks for the comment. For Fig. 1-5, we added “one-way ANOVA and Tukey's multiple comparison” into the figure legend to explain the statistic method for the results. As for Fig. 6, “Pearson correlation” is in the title of this figure to indicate the nature of our correlation analysis and statistical analysis.
- “In the Result section in points 4.2, 4.3 and 4.4 information is presented regarding the correction of specific quantities or results, and there is no specific reference to the specific values of these correlations. These considerations are not supported by any correlation coefficient values or ANOVA statistical analysis results. It is necessary to supplement this point. After all, the authors write that they have carried out such an analysis, but the results of this analysis are missing in the article.”
Reply: Sections 4.2, 4.3 and 4.4 belong in the Discussion section, not the Results section of this manuscript. Therefore, we did not repeat the narratives on the correlations among the measured parameters. For example, the correlations of proximate analysis composition, lipid oxidation, and antioxidants with protein oxidation in SBMs have been provided in section 3.5 (Line 285). Instead, we focused more on explaining the causes behind different oxidation levels between MSBM and SSBM and the significance of these differences in these sections.
- “This section contains the basic conclusions of the research, but should be revised after slightly supplementing the article with the results and comments on the statistical analysis.”
Reply: Thanks for the comment. We have revised the Conclusions section by including more specific statements on the results based on the statistical analysis.
- “References should be prepared in accordance with the guidelines. There are minor errors such as a minor error in the name of the journal: e.g. it should be: Poultry Science ([9]), so please check all literature items. Journals names should be in capital letters, and there are different styles, etc., e.g. compare [5] and [6]. The authors sometimes use the full name of the journal, and sometimes the abbreviation, e.g. compare [13] and [14].”
Reply: The references have been revised as indicated.
Reviewer 2 Report
This manuscript is very well organised. It is really a very interesting work with excellent indicators to assess quality.
Author Response
--- Responses to Reviewer 2’s comments
“This manuscript is very well organized. It is really a very interesting work with excellent indicators to assess quality.”
Reply: Thank you!
Reviewer 3 Report
I am very grateful you for the invitation to review manuscript antioxidants-2489358 by Zhang and coauthors "Counteracting roles of lipidic aldehydes and phenolic antioxidants on soy protein oxidation defined by a chemometric survey of solvent and mechanically extracted soybean meals”. In the present study was evaluated the consumer's understanding of the expiration dates indicated on packing; identify consumer behavior toward expired food and know the consumer's perception of the health risks of consuming expired food. The work is interesting but needs adjustments to increase the quality of the material.
Comments:
- Abstract, Line 11-13: Specify in more detail the problems related to oxidation for animal feed.
- Abstract: Please include the objective directly in the abstract.
- Please indicate a brief and better step-by-step about the work including the parameters and conditions used.
- Line 16: Was the difference better or bigger?
- The conclusion of the abstract does not respond to the proposed objectives. The conclusion must be reviewed and readjusted.
- Line 30: Change the repeated keywords by different words from the title.
- Line 33-36: Introduction: Indicate soybean production and the amount intended for animal feed.
- Line 61: Better explain the negative effect of antinutrients.
- Lines 66-69: The negative impacts of the formation of these components are not explained.
- Introduction: The presentation of the study objectives is not clear in this item. Please insert a sentence about the objectives clearly.
- Line 93: Authors can include calculation of carbohydrates by difference and caloric value based on the concentration of each component.
- Line 190: Standardize terminology throughout the text. This is the only point where the raw material is referred to as “3.1. Biomass”.
- Line 298: Specify the products formed and which have problems.
- Discussion: Deepen the discussion regarding the difference in material, especially the difference in composition in relation to the year in which the material was obtained.
- Discussion: It is generally well written and detailed.
Author Response
--- Responses to Reviewer 3’s comments
- “- Abstract, Line 11-13: Specify in more detail the problems related to oxidation for animal feed.”
Reply: We added a brief statement “its nutritional value can be compromised by protein oxidation” into the Abstract. More details on the problems related to the oxidation of feed ingredients oxidation were described in the Introduction section (Line 53-71).
- - Abstract: Please include the objective directly in the abstract.
Reply: We revised the Abstract to state the objective of the current study as follows “… to determine the extent of protein oxidation and its correlation with soybean oil extraction methods and non-protein components.” (Line 15-16).
- “- Please indicate a brief and better step-by-step about the work including the parameters and conditions used.”
Reply: Our approach for writing Materials and Method in this manuscript was to provide step-by-step details on some special and less common methods (e.g., protein carbonyl content and aldehydes), while mainly listing the references with technical details for more common and widely accepted methods (e.g., proximate analysis).
- “- Line 16: Was the difference better or bigger?”
Reply: The current study does not aim to define whether the observed differences between SSBM and MSBM are better or worse, but we are more interested in understanding the correlations among measured parameters as well as their implications in soybean meal production and animal feeding.
- “- The conclusion of the abstract does not respond to the proposed objectives. The conclusion must be reviewed and readjusted.”
Reply: We believe that we provided adequate concluding statements in the abstract as follows:
---The direct conclusion of this study was described as “Overall, soybean oil extraction methods, together with other factors such as enzyme treatment and environmental conditions, can significantly affect proximate analysis composition, protein and lipid oxidation status, and the antioxidant profile of SBM. Lipidic aldehydes and phenolic antioxidants play counteracting roles in the oxidation of soy protein.” (Line 24-28)
--- Moreover, a major practical application of this conclusion was stated as “The range of protein carbonyl content measured in this study could serve as a reference to evaluate the protein quality of SBM from various sources used in animal feeds”. (Line 28-29)
- “- Line 30: Change the repeated keywords by different words from the title.”
Reply: The only key word that was repeated was “oxidation”, which applies to two separate processes for protein oxidation and lipid oxidation and therefore should not be considered as a repetition.
- “- Line 33-36: Introduction: Indicate soybean production and the amount intended for animal feed.”
Reply: The quantities of SBM production and its usage in animal feeding have been provided in revised Introduction as suggested (Line 33-34).
- “- Line 61: Better explain the negative effect of antinutrients.”
Reply: The inclusion of “antinutritional native soy proteins” in the Introduction was for explaining the functions of thermal treatments in SBM preparation. Antinutrients in SBM was not the focus of this study. Therefore, we consider that the current sentence “Normally, controlled heating is purposely used in SBM processing to denature antinutritional native soy proteins, especially trypsin inhibitors, in raw soybeans to improve their nutritional value [13, 14]” (Line 60-62) is sufficient for the purpose.
- “- Lines 66-69: The negative impacts of the formation of these components are not explained.”
Reply: In the Introduction, the negative impacts of these lipid oxidation products were stated as follows:
--- “The consequences of these pro-oxidation events are the peroxidation of unsaturated fatty acids and the formation of lipid oxidation products (LOPs) [15,16]. Among the diverse LOPs, reactive aldehydes can covalently bond with amino and sulfhydryl groups of amino acids, forming carbonylated soy proteins with decreased solubility and digestibility [20, 21].” (Line 67-71)
- “- Introduction: The presentation of the study objectives is not clear in this item. Please insert a sentence about the objectives clearly.”
Reply: We modified the introduction description of objectives as suggested.
- “- Line 93: Authors can include calculation of carbohydrates by difference and caloric value based on the concentration of each component.”
Reply: Soybean meal contains about 40% carbohydrates, but estimating the caloric value of proximate components is not relevant to the focus of our study, and would vary among animal species. Instead, we focused on components that could contribute more to protein and lipid oxidation in the current study, such as ether extract and moisture.
- “- Line 190: Standardize terminology throughout the text. This is the only point where the raw material is referred to as “3.1. Biomass”.”
Reply: Thanks for the suggestion. We have replaced “Biomass” with “Proximate Analysis”.
- “- Line 298: Specify the products formed and which have problems.”
Reply: We have now included “such as carbonylated protein, crosslinked peptides, and lipidic aldehydes” (Line 304-305) to explain the forms of oxidized proteins and lipids in feed ingredients that can cause negative effects in animal feeding.
- “- Discussion: Deepen the discussion regarding the difference in material, especially the difference in composition in relation to the year in which the material was obtained.”
Reply: Discussion on the differences between SSBM and MSBM in the protein oxidation, lipid oxidation, gross composition, and their correlations with protein oxidation are presented in sections 4.1. 4.2, 4.3, and 4.4, respectively. For example, “4.1 Protein Oxidation Status in SSBM and MSBM” focuses on the principle of protein oxidation, differences in protein carbonyl content between SSBM and MSBM, practical application of the comprehensive chemometric survey, and effects of processing and storage on protein oxidation. As for the differences between 2021 SSBM and 2020 SSBM samples, we also explore the potential influences of climate (Line 374-378). Therefore, we consider we made our best efforts in the discussion on these topics.
- “- Discussion: It is generally well written and detailed.”
Reply: Thank you.